# Quantifying the Generalization Gap in Seizure Detection: A Large-Scale Empirical Benchmark via the SzCORE Challenge

**Jonathan Dan** [1]  **Amirhossein Shahbazinia** [1]  **Christodoulos Kechris** [1]  **David Atienza** [1]

## Abstract

Reliable automatic seizure detection from long-term electroencephalogram recordings (EEG) remains an unsolved challenge, as current models often fail to generalize across patients or clinical settings. Manual EEG review still is the standard of care, highlighting the need for robust models and standardized evaluation. The current literature often reports high efficacy, yet these models frequently fail when deployed to unseen patient populations. To rigorously assess this generalization gap, we conducted a large-scale empirical study evaluating 28 state-of-the-art algorithmic architectures, ranging from classical feature engineering to modern Deep Learning. These algorithms were collected by organizing a competition. A strictly held-out private dataset of continuous EEG recordings from 65 subjects, totaling 4,360 hours of data, was utilized to evaluate algorithm performance. Expert neurophysiologists annotated these recordings, establishing the ground truth for seizure events. Algorithms were evaluated using event-based metrics from the SzCORE framework, including sensitivity, precision, F1-score, and false positive rate per day. Results revealed significant performance variability among state-of-the-art approaches, with the top F1 score of 32% (sensitivity 37%, precision 29%), highlighting the persistent difficulty of this task for current machine learning methodologies. Our analysis uncovered a discordance between peak performance and population-level stability. The algorithms achieving the highest aggregate F1-scores did not achieve the most consistent ranking across subjects, indicating high performance variance and susceptibility to failure on outlier patients. This independent evaluation also exposed

a notable gap between self-reported efficacies and hold-out performance, underscoring the critical need for standardized, rigorous benchmarking in developing clinically viable ML models. A comparison with previous challenges and commercial systems indicates that the best algorithm in this study surpassed prior methods. Critically, the evaluation infrastructure transitions into a continuously open benchmarking platform, fostering reproducible research and accelerating the development of robust seizure detection algorithms by allowing ongoing submissions and integration of additional private datasets. Clinical centers can also adopt this platform to evaluate seizure detection algorithms on their EEG data using a standardized, reproducible framework.

## 1. Introduction

Automated seizure detection is an active and critical area of machine learning research, particularly driven by the increasing use of ambulatory and long-term electroencephalography (EEG) monitoring in patients with epilepsy. These monitoring techniques, such as those used in epilepsy monitoring units (EMU) and home-based EEG, aim to provide continuous, real-time insights into a patient's condition, improving diagnostic accuracy and enabling timely medical interventions. Real-time seizure detection can significantly benefit clinical practice by providing immediate alerts to healthcare professionals during in-hospital monitoring (Kamitaki et al., 2019). Meanwhile, offline detection helps reduce clinicians' workload and detect subtle, often difficult-to-identify seizures (Baumgartner and Koren, 2018).

Despite significant advancements in the development of automated methods, algorithms often struggle to generalize across different patients, clinical settings, and seizure types. As a result, manual EEG review by clinicians remains the gold standard, underscoring the urgent need for robust, generalizable algorithms that can be deployed in real-world clinical environments. This gap in the state of the art presents a significant opportunity for innovation in machine learning and medical applications.

[1]Embedded Systems Laboratory, EPFL, Lausanne, Switzerland. Correspondence to: Jonathan Dan <jonathan.dan@epfl.ch>.

One critical obstacle hindering progress in this field is the lack of standardized evaluation practices, which complicates direct comparisons across studies and prevents the community from identifying the best-performing algorithms. Many recent reviews of seizure detection methods provide valuable overviews of existing work but fail to compare state-of-the-art seizure detection algorithms directly. Miltiadous et al. (Miltiadous et al., 2023) conducted a systematic review according to the PRISMA guidelines. They summarize the performance metrics reported by the authors for 89 studies in the Bonn dataset (Andrzejak et al., 2001), 20 studies in the Physionet CHB-MIT Scalp EEG database (Goldberger et al., 2000; Shoeb, 2009), 16 in other datasets, and 65 in combinations of datasets. The results are organized by dataset. Shoebi et al. (Shoeibi et al., 2021) focused on categorizing deep learning methods. They report the accuracy of the authors' findings for 136 studies grouped by deep-learning technique. Similar approaches have been developed in other recent reviews (Supriya et al., 2023; Siddiqui et al., 2020; Baumgartner and Koren, 2018). Although these reviews provide a comprehensive overview of the recently published work with a detailed analysis of the datasets used for training and evaluation, along with categorization of the most common machine learning techniques, they do not allow a direct comparison of performance due to heterogeneity in the datasets, evaluation metrics, and evaluation methodology. Most reviews acknowledge this limitation.

To address these challenges, Dan et al. developed Sz-CORE (Dan et al., 2024), a framework that standardizes the validation methodology of seizure detection algorithms and enables direct comparison across studies. SzCORE proposes guidelines on EEG datasets, evaluation methodologies, and performance metrics. It defines the file format and data organization of EEG datasets. These follow a specific format that complies with the BIDS-EEG (Gorgolewski et al., 2016; Pernet et al., 2019) and HED-SCORE (Hermes et al., 2024) specifications. SzCORE defines event-based scoring to compute performance metrics. Event-based scoring compares the labels of an event (seizure) based on the overlap between the reference (ground truth) and the hypothesis (algorithm output) to count True Positives (TP), False Positives (FP) and False Negatives (FN). These counts are then used to compute four performance metrics: sensitivity, precision, F1-score, and false positive rate per day (FP/24h). Algorithms evaluated with SzCORE can be directly compared as performance metrics are computed using the same methodology. By providing this framework, SzCORE aims to accelerate iterative improvements in machine learning models for seizure detection, allowing researchers to more reliably build upon prior art.

A standardized evaluation framework, such as SzCORE, provides a consistent methodology for comparing algorithms, making it an ideal basis for self-evaluation. However, it does not provide a submission and evaluation pipeline for benchmarking algorithms on a private dataset.

This paper presents a large-scale empirical study investigating the limits of generalization in automated seizure detection. To ensure a diverse and representative evaluation of the state-of-the-art, we compiled a library of algorithms through a community-wide benchmarking initiative organized in conjunction with the 2025 International Conference on Artificial Intelligence in Epilepsy and Neurological Disorders held in Breckenridge (Colorado, USA) from March 3 to 6, 2025. The SzCORE framework was used for a standardized evaluation. This work represents the first large-scale direct comparison of seizure detection algorithms using a standardized evaluation methodology on a substantial private dataset. Furthermore, by analyzing the characteristics of submitted algorithms alongside their performance, we aim to shed light on effective strategies and persistent hurdles in developing automated seizure detection systems. The challenge's source code and evaluation platform will remain online as a public service.

This initiative provides a crucial and persistent resource for the machine learning community, promoting transparency, reproducibility, and continuous progress in this vital area of medical AI, establishing a new paradigm for benchmarking seizure detection algorithms on private datasets.

We describe the algorithm collection in Section 2, the dataset (Section 3), the evaluation procedure (Section 4) and the algorithmic contributions (Section 5). We analyze the results in terms of event-based F1-score and computational cost (Section 6). We compare them to self-reported results, previous challenges, and commercial software and discuss the limitations of the challenge in the discussion (Section 7).

## 2. Algorithm Collection

To rigorously assess the generalization capabilities of current machine learning methods, we defined a standardized seizure detection task requiring the automated identification of seizure onset and duration in continuous, long-term (multi-day) EEG recordings from the Epilepsy Monitoring Unit (EMU).

To ensure fair and reproducible comparison across diverse architectures, all algorithms were required to adhere to strict input/output specifications. Long-term continuous EEG signals from the EMU were used as input data. The recordings were stored in `.edf` files. EEG acquisition comprised 19 electrodes in the international 10-20 system in a referential, common average montage (Schomer and Lopes da Silva, 2017) (see Section 3 for details on standardization). The recordings were sampled at 256 Hz. Each file was guaranteed to last at least 1 minute, with most files lasting approximately an hour.

Algorithm outputs were standardized to the HED-SCORE compliant format (Dan et al., 2024), which uses a tab-separated value `.tsv` file as an output. This text file uses a tab as a delimiter to separate the different columns of information, each row representing one seizure event. Each annotation file is associated with a single EEG recording.

To compile a representative cross-section of the state-of-the-art, we solicited algorithmic submissions via a community-wide call for participation in the form of a challenge that was organized in conjunction with the 2025 International Conference on Artificial Intelligence in Epilepsy and Neurological Disorders held in Breckenridge (Colorado, USA). Algorithms were submitted between 1 December 2024 and 16 February 2025.

Contributors were required to submit pre-trained algorithms packaged as isolated Docker images. This containerization strategy ensured that the evaluation was fully reproducible and blind to the specific internal mechanics of the models (allowing for both open-source and proprietary submissions). Alongside the Docker image, contributors provided an abstract detailing their methodology and training data. To reflect realistic deployment scenarios where models are pre-trained on available data before clinical deployment, algorithms were permitted to use any combination of public and private training datasets. We provided code to convert the major public datasets to SzCORE-formatted versions. Specifically, the Physionet CHB-MIT Scalp EEG Database, the Physionet Siena Scalp EEG Database, and the TUH EEG Sz Corpus were supported to facilitate baseline development and ensure consistent data handling.

## 3. Evaluation dataset

The evaluation dataset comprised previously collected recordings from 65 patients (resulting in 398 annotated seizures) in the EMU of the Filadelfia Danish Epilepsy Centre in Dianalund, between January 2018 and December 2020. Data had been recorded with the NicoletOne™ v44 amplifier. The dataset included patients who had at least one seizure during the hospital stay with a visually identifiable electrographic correlate to clinical seizures recorded on video. The total recording duration amounted to 4'360 hours. Subject recordings ranged from 18 hours to 98 hours. Most participants were adults, with a median age of 34 years (range: 5 to 66 years). Eight children were included in the study. Seizures had been independently annotated by three board-certified neurophysiologists with expertise in long-term video-EEG monitoring. In case of disagreement, a ground-truth label was obtained after a consensus discussion between the experts, ensuring high-fidelity ground truth for the evaluation. The statistics of the dataset are illustrated in Figure 1. Data were anonymized and converted to a BIDS-compatible (Gorgolewski et al., 2016; Pernet et al.,

2019) format using the `epilepsy2bids` Python library. This library, which is publicly available on GitHub, was specifically adapted to support the Filadelfia dataset. The conversion standardizes the EEG channels' number, naming, and order to the 19 channels of the 10-20 system. They are re-referenced to a common average montage and uniformly sampled at 256 Hz. Open-source submissions were also evaluated on three public datasets and one extra private datasets. The additional evaluation datasets are Physionet CHB-MIT v1.0.0 (Shoeb, 2009; Goldberger et al., 2000), Physionet Siena v1.0.0 (Detti, 2020; Detti et al., 2020; Goldberger et al., 2000), TUH Seizure Corpus v2.0.3 and KU Leuven SeizeIT1 v1.0.0 (Vandecasteele et al., 2020; Chatzichristos and Claro Bhagubai, 2023).

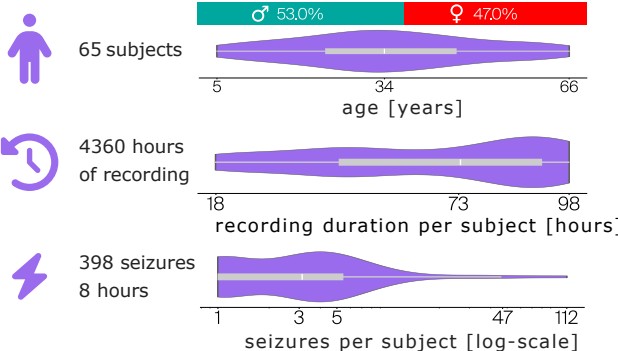

*Figure 1.* Distribution of the data in the Filadelfia dataset.

## 4. Evaluation procedure and metrics

To ensure a rigorous and reproducible assessment of generalization performance, we established a containerized, blind evaluation protocol executed on high-performance computing infrastructure.

**Reproducible Execution Environment**  To eliminate dependency-related discrepancies, all algorithms were encapsulated as isolated Docker images. This containerization ensured compatibility across computing environments and allowed for the strict separation of training and testing phases. All algorithms were evaluated in a blind setting with inference performed on the held-out evaluation dataset (Section 3) within an isolated environment, preventing any potential data leakage or test-set optimization.

**Computational Infrastructure**  Evaluations were conducted on an institutional High-Performance Computing (HPC) platform utilizing a Container-as-a-Service (CaaS) model. The infrastructure provided access to over 400 GPUs of different types. In the context of this study, A100 NVIDIA GPU nodes with $2 \times$ CPU AMD EPYC 7543 32c/64t, 1 TB RAM, and $8 \times$ NVIDIA HGX A100-SXM4-80GB NVLINK were used for GPU workloads. V100

NVIDIA GPU nodes with $2 \times$ INTEL Gold 6240, 384 GB RAM and $4\times$ NVIDIA v100-SXM2-32GB NVLINK were used for CPU-only loads. The computing platform provides detailed reports on the total GPU hours, CPU hours, and average RAM usage per hour used to execute a computational job. The GPU hours are calculated based on reservation time, whereas CPU and RAM usage are based on actual utilization. We manually determined the execution parameters by evaluating algorithms on a small subset of the dataset; a pragmatic approach given the diversity of submitted computational requirements. This determined the GPU reservation requirements and the number of files to process in parallel within each Docker container. We further split the inference across multiple Docker instances when required to accelerate execution.

**The performance metrics**   Reported metrics include sensitivity, precision, F1-score, and false positives per day. They are computed using the `timescoring` library, which is described in SzCORE (Dan et al., 2024). We prioritized event-based scoring as the primary evaluation mode because it better reflects clinical decision-making by treating each seizure as a discrete diagnostic event. Event-based scoring relies on overlap: if a reference (ground-truth) event and a hypothesis (algorithm) event overlap, the detection is correct (True Positive); if a hypothesis event does not overlap with any reference event, it is considered a false detection (False Positive). The following event-based scoring parameters (default values from the SzCORE framework) are used in this challenge:

- Minimum overlap: between the reference and hypothesis for a detection. We use any overlap, however short, to enhance sensitivity.

- Pre-ictal tolerance: tolerance with respect to the onset of an event that would count as a detection. We use a 30-second pre-ictal tolerance.

- Post-ictal tolerance: tolerance with respect to the end time of an event that would still count as a detection. We use a 60-second post-ictal tolerance.

- Minimum duration: between events resulting in merging events that are separated by less than the given duration. We merge events separated by less than 90 seconds, which corresponds to the combined pre- and post-ictal tolerance.

- Maximum event duration: splitting events longer than the given duration into multiple events: We split events longer than 5 minutes.

Scores are computed per subject by adding the true positives, false positives and false negatives of the individual recordings to calculate sensitivity, precision, F1-score and false positives per day. These scores are then averaged with the arithmetic mean over the different subjects. The `szcore-evaluation` library was used to compute performance metrics across the entire dataset. If an algorithm fails to produce a `.tsv` seizure annotation file for a given `.edf` file, it is considered that the algorithm did not predict any seizures for that file. No detections for a subject results in an undefined precision, which is set to zero when computing the mean.

**The primary ranking criterion**   was the event-based F1-score, as it balances seizure detection rate (sensitivity) against precision. It is computed as the harmonic mean of sensitivity and precision. This metric represents the predictive performance of an algorithm. This metric is particularly robust for imbalanced classes, such as epileptic seizure detection where positive events are rare.

**Statistical Analysis**   To rigorously compare algorithmic performance across the 65 subjects, we employed non-parametric statistical testing suitable for multiple classifiers evaluated on a common dataset (Demšar, 2006). First, we applied the **Friedman test** to detect statistically significant differences in the distributions of event-based F1-scores across all algorithms. This omnibus test evaluates the null hypothesis that all algorithms perform equivalently in terms of ranking. Following a significant omnibus result ($p < 0.05$), we conducted post-hoc pairwise comparisons to identify the top-tier algorithms. We utilized the Wilcoxon signed-rank test with Holm-Bonferroni correction to control the family-wise error rate when comparing the top-ranked algorithm against all other submissions. We also computed the Nemenyi test for all-pairwise comparisons to generate a critical difference diagram. This visualizes groups of algorithms that are not statistically distinguishable in terms of average rank. Finally, to quantify the magnitude of performance differences beyond significance, we calculated Cliff's Delta ($\delta$), a non-parametric effect size measure. We interpret effect sizes as negligible ($|\delta| < 0.147$), small ($< 0.33$), medium ($< 0.474$), or large ($> 0.474$).

## 5. Evaluated algorithms

To ensure a comprehensive audit of the field, we analyzed 28 distinct algorithmic executions (derived from 20 unique architectural approaches). These models represent a broad cross-section of current methodologies, ranging from classical feature engineering to state-of-the-art deep learning. Algorithms process the input signal in fixed-duration windows, which slide with a predefined step to the next input. The window duration ranged from a few seconds to a few minutes. The algorithm processing can typically be described in three steps: 1. pre-processing the input windows - filtering

and sometimes extracting signal features, 2. processing the inputs with a machine-learning model, 3. post-processing the model outputs.

**Pre-processing** Most algorithms included a pre-processing step comprising a band-stop and / or band-pass filter (Sanei and Chambers, 2007). Some algorithms extracted features of the EEG after filtering. Four algorithms used the short-time Fourier transform (STFT) of the EEG as input to the model. The *Gradient Boost v1, v2, v3* algorithms extracted linear and non-linear connectivity features. The *Random Forest* algorithm combined time features with frequency ones. The *NE Illusion* algorithm included a channel selection step. Similarly, the *STORM* algorithm removed low-quality EEG channels.

**Processing** Deep learning methods were used in 17 submissions, with 13 algorithms processing inputs in the time domain, while the four others processed STFT inputs.

**Post-processing** included several variations to smooth the model output. Implementations included moving average or median filters, merging nearby seizure events, discarding short ones, and thresholding predictions.

Participants were allowed to submit multiple versions of their method, resulting in five methods with multiple versions, ranging from two to three versions each. Variations between versions included:

1. Changes in the classification threshold observed in *Gradient Boost*, *HySEIZ* and *S4Seizure*.

2. Changes in the model architecture observed in *HySEIZ*.

The steps and methods used by each algorithm are detailed in Appendix A (Table 2). The relevant information for each method was extracted from the accompanying abstract, provided by the development team during the submission stage, [1] and their corresponding GitHub repository. The information was verified with the algorithm developers. The datasets used to develop each model are detailed in Appendix B (Table 3).

# 6. Results

Of the thirty algorithms submitted to the study, twenty-eight were successfully evaluated on the full held-out dataset (two were excluded due to runtime failures). The event-based scores are presented in Table 1. The winning submission (*Sz*

---

[1]The abstracts and GitHub repositories can be accessed in the algorithm information page available for each algorithm in the benchmark website `https://epilepsybenchmarks.com/challenge`

*Transformer*) (Wu et al., 2025) has an F1-score of 32%, a sensitivity of 37%, a precision of 29%, and produces an average of 1.34 false positives per day. The top-5 submissions have a sensitivity ranging from 30%-58%, a precision of 19%-29%, and produce between 1.34 and 14 false positives per day. The sensitivity as a function of precision is shown for all submissions in Figure 2. In the appendix D, we also report the event-based F1-score of all the open-source submissions on four additional datasets.

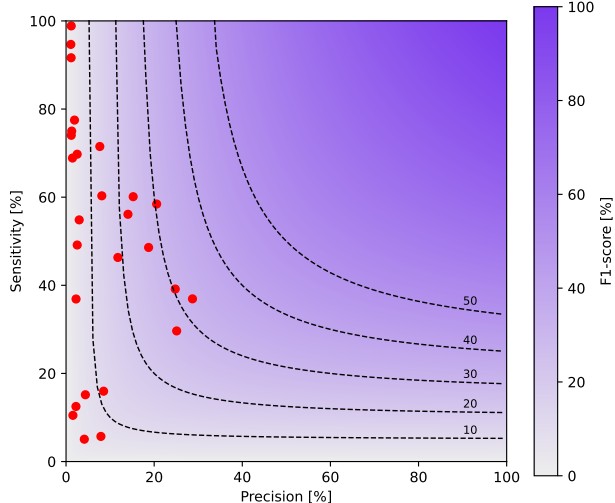

*Figure 2.* Sensitivity as a function of precision for the • algorithms submitted in the challenge. The background is shaded according to the F1-score, with dashed lines indicating iso-F1 score.

The Friedman test confirmed significant performance differences across the field ($\chi^2 = 433.55, p < 0.001$). While *Sz Transformer* achieved the highest aggregate F1-score, *HySEIZa v1* demonstrated superior stability across the population, resulting in the best mean rank.

Post-hoc Wilcoxon signed-rank tests with Holm-Bonferroni correction (Table 7) demonstrate that there is no statistically significant difference ($p_{corr} > 0.05$) between the top-ranked *HySEIZa v1* and the F1-leader *Sz Transformer*. The effect size between these two contrasting architectures is negligible (Cliff's $\delta = 0.01$).

Seven leading architectures are statistically indistinguishable from the rank-leader as shown in the critical difference diagram (Figure 4). Conversely, classical baselines like *Random Forest* and *Channel-adaptive classifier* were significantly outperformed with large effect sizes (Cliff's $\delta > 0.6$).

The agreement between the algorithms is shown in Figure 5. It shows that a single seizure event was missed by all algorithms, and that the seizure with the most detections was detected by 26 out of the 28 algorithms. Notably, the figure also shows low agreement between false positives,

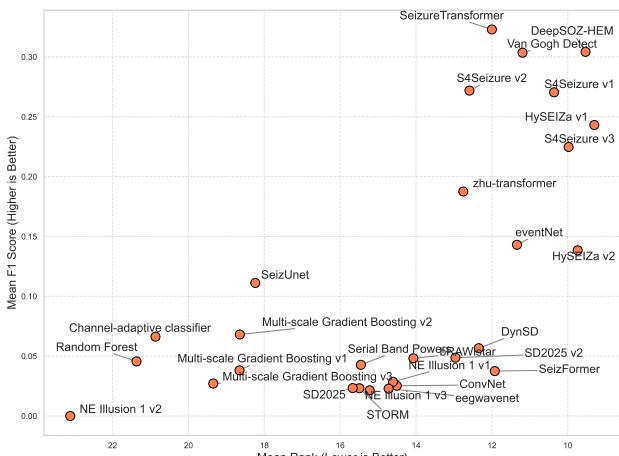

*Figure 3.* Trade-off analysis between mean Rank and F1 Score across all models. Top-performing models are located in the upper-right quadrant (low rank, high F1 score). Note that the x-axis is inverted to reflect that lower ranks represent superior performance.

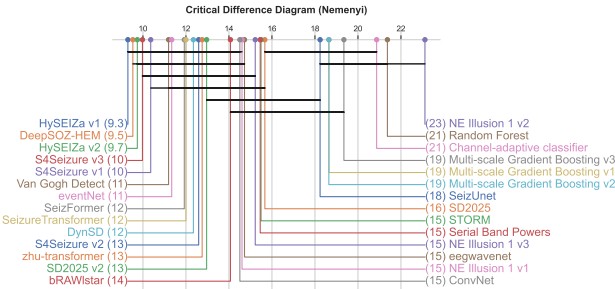

*Figure 4.* Post-hoc Nemenyi test results for pairwise comparisons of model performance. Solid black lines represent algorithms with ranking that are statistically not different.

with less than 25% of the algorithms agreeing on 91% of the false detections. This suggests that false positives are largely idiosyncratic to individual algorithms rather than stemming from universally misleading artifacts in the data. We also found that hard-to-detect seizures (<25% detection) are substantially shorter (median 48 s vs. 118 s for easy seizures (>75% detection), suggesting that brief ictal patterns often fall below the processing windows of current algorithms, which range from 1 to 600 s in submissions. Hard seizures are also less frequently nocturnal (22.5%) than easy ones (27.8%), indicating that time-of-day is not a primary failure mode. At the subject level, 15 of the 65 subjects (23%) recorded $F_1 = 0$ in all top-5 algorithms simultaneously, pointing to patient-specific electrographic patterns not captured by any submission, rather than a systematic framework failure. The one seizure missed by all 28 algorithms occurred during the day.

The difference between the performance reported by algorithm developers on their self-selected datasets and the event-based F1-score on the challenge evaluation dataset

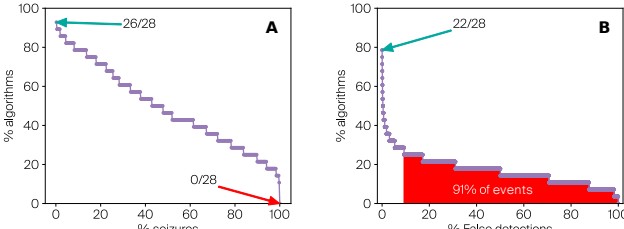

*Figure 5.* Percentage of algorithms that detect a percentage of events. Panel **A** shows the true positives, and panel **B** shows the false positives.

is represented in Figure 6. This comparison starkly illustrates that nearly all algorithms strongly overestimated their performance compared to the results obtained on this independent test set, highlighting a critical challenge in the field regarding generalizability and the necessity of standardized, independent benchmarking.

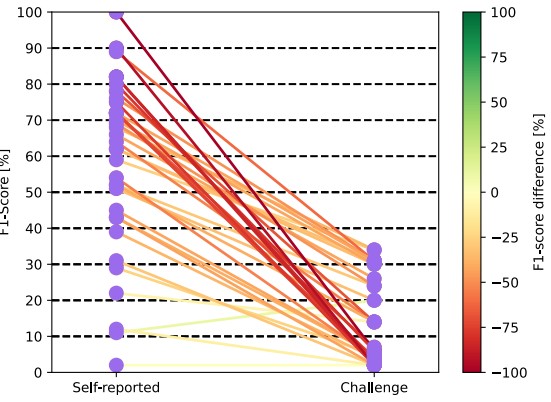

*Figure 6.* F1-score self-reported by the algorithm developers versus the event-based F1-score obtained in this challenge. The difference in F1-score is represented by the color of the line.

Evaluating all the algorithms required 334 Nvidia A100 GPU hours, 97'885 CPU hours (76'668 hours of Intel Gold 6240 and 21'217 hours of AMD EPYC 7543 32c/64t) and 403'933 GB of RAM. The computing resources are detailed in Appendix C (Table 4).

## 7. Discussion

This study aimed to rigorously evaluate the current state of machine learning algorithms for automatic seizure detection in long-term EEG recordings from the EMU, leveraging a large, private dataset and the standardized SzCORE evaluation framework. With 30 submissions from 19 teams, this initiative represents a significant, contemporary cross-section of methodologies in the field.

| Algorithm | F1 | Sens. | Prec. | FP/day |
|---|---|---|---|---|
| Sz Transformer | 32 | 37 | 29 | 1 |
| Van Gogh Detect | 30 | 39 | 25 | 3 |
| DeepSOZ-HEM | 30 | 58 | 21 | 14 |
| S4Seizure v2 | 27 | 30 | 25 | 2 |
| S4Seizure v1 | 27 | 49 | 19 | 7 |
| HySEIZa v1 | 24 | 60 | 15 | 13 |
| S4Seizure v3 | 22 | 56 | 14 | 13 |
| zhu-transformer | 19 | 46 | 12 | 24 |
| HySEIZa v2 | 14 | 72 | 8 | 29 |
| eventNet | 14 | 60 | 8 | 20 |
| SeizUnet | 11 | 16 | 9 | 4 |
| Channel-adaptive | 7 | 6 | 8 | 1 |
| Gradient Boost v2 | 7 | 15 | 4 | 6 |
| DynaSD | 6 | 55 | 3 | 37 |
| Random Forest | 5 | 5 | 4 | 1 |
| SD2025 v2 | 5 | 70 | 3 | 86 |
| bRAWlstar | 5 | 49 | 3 | 46 |
| Gradient Boost v1 | 4 | 12 | 2 | 12 |
| Band Powers | 4 | 37 | 2 | 50 |
| SeizFormer | 4 | 77 | 2 | 83 |
| NE Illusion 1 v1 | 3 | 69 | 1 | 158 |
| Gradient Boost v3 | 3 | 10 | 2 | 22 |
| ConvNet | 3 | 75 | 1 | 163 |
| NE Illusion 1 v3 | 2 | 95 | 1 | 280 |
| STORM | 2 | 99 | 1 | 290 |
| eegwavenet | 2 | 92 | 1 | 237 |
| SD2025 | 2 | 74 | 1 | 188 |
| NE Illusion 1 v2 | 0 | 0 | | 0 |

*Table 1.* Event-based scores

The organization of challenges is a recognized catalyst for innovation within machine learning, famously demonstrated by events like the ImageNet Large-Scale Visual Recognition Challenge (Russakovsky et al., 2015), which was pivotal in showcasing the power of Convolutional Neural Networks. Within epilepsy research, previous challenges have targeted diverse aspects, from intracranial EEG (upe, 2014) to optimizing electrode configurations (neu, 2020) and wearable EEG (Chatzichristos et al., 2023). Our challenge builds upon these efforts, uniquely focusing on the prevalent 19-channel scalp EEG setup from EMUs. The scale of participation (28 successfully evaluated algorithms) and the performance of the top submissions (F1-score of 32%, sensitivity 37%, precision 29% at 1.34 FP/day) indicate an advancement over prior scalp-EEG challenges. For instance, the leading algorithm in the Neureka challenge achieved 12.37% sensitivity at 1.44 FP/day on a different dataset and scoring methodology (Shah et al., 2018), suggesting tangible progress in algorithmic capabilities.

When benchmarked against commercial systems as evaluated by Koren et al. (Koren et al., 2021) (who used a com-parable, albeit distinct, dataset and slightly more permissive event-scoring), our top-performing algorithms demonstrate competitive F1-scores (e.g., Encevis 1.7 achieved 44.1%). This suggests that research-driven algorithms are approaching, and in some aspects matching, the performance of established commercial solutions. However, it's crucial to note that commercial systems often operate at a different point on the sensitivity-precision curve, typically prioritizing higher sensitivity (68-81%) at the cost of increased false positives, a trade-off reflecting specific clinical needs.

The choice of the event-based F1-score as the primary ranking criterion in this challenge encouraged algorithms that balance between sensitivity and precision. Our results, particularly the analysis of multiple submissions from the same team with varied thresholds (e.g., *S4Seizure*), illustrate the critical impact of operating point selection. However, the challenge format did not allow participants to fully explore the sensitivity-precision curve, which might limit the clinical utility of the winning solution. An important direction for future research is to develop methods that can robustly optimize this trade-off or allow clinicians to easily tune it post-hoc based on their specific requirements, without needing access to the original evaluation dataset. Analysis of the ranking showed, higher average F1-score does not translate to better ranking across the patient population. This is due to the high-variability of some of the algorithms that detect few events. These algorithms can obtain very high F1-scores or very low scores when the seizures as missed. This high variability leads to a lower average ranking. Statistical testing on the rank shows that some of the leading algorithms are indistinguishable from the rank-leader in this dataset.

A striking finding from this challenge is the significant discrepancy between the self-reported performance of algorithms on developers' chosen datasets and their actual performance on our independent evaluation dataset (Figure 6). This observation, common in machine learning but particularly critical in medical AI, underscores the profound impact of dataset characteristics and evaluation methodologies. The Filadelfia dataset, with its inclusion of pediatric patients and data from portable EEG systems allowing patient mobility, likely presents a more complex and realistic distribution of signals and artifacts than many curated public datasets. This "generalization gap" highlights the insufficiency of relying solely on self-reported metrics and the imperative need for independent, standardized benchmarking on diverse, clinically representative data that moves beyond performance comparisons to include a deeper investigation into the common architectural principles of successful models and a root-cause analysis of their failures. Future initiatives could further enhance generalizability assessment by incorporating multi-centric evaluation datasets, reflecting the wider variability in EMU equipment and protocols.

A core contribution of this work, extending beyond the immediate results of the challenge, is the transition of the entire evaluation infrastructure into a continuously open benchmarking platform with submissions as pull-requests on a public GitHub repository: `https://github.com/esl-epfl/szcore`. This initiative directly addresses the aforementioned critical need for standardized, reproducible, and ongoing assessment in the field of seizure detection. By allowing continuous submissions, this platform will serve as a living benchmark, enabling researchers to:

- Evaluate new algorithms against a consistent, private, and expertly annotated dataset under a unified scoring framework (SzCORE).

- Objectively track the progress of the field over time, mitigating the issue of self-reporting bias.

- Foster transparency and reproducibility, as methods can be directly compared.

- Potentially integrate additional private datasets from other clinical centers under the same standardized evaluation pipeline, broadening the benchmark's scope and robustness.

This open platform is designed to become a vital community resource, significantly lowering the barrier to rigorous evaluation and accelerating the iterative development and validation of robust seizure detection algorithms. For the ML community, this provides a concrete tool and a new paradigm for benchmarking in a challenging medical domain, promoting best practices in ML evaluation.

Finally, while the top-performing algorithms in this challenge show promise, their direct translation to clinical practice requires further steps. The current F1-scores, while competitive, may still fall short of clinical expectations for sensitivity in many scenarios. Further algorithmic refinement is needed, potentially focusing on achieving much higher sensitivity, even if it incurs a higher false positive rate that clinicians might manage with appropriate alert systems and review interfaces. The standardization efforts inherent in SzCORE and this benchmarking platform can facilitate this transition by ensuring that new algorithms can be seamlessly integrated into EEG review software that adheres to these established input/output formats. This framework supports the development of more sophisticated clinical decision support tools, ultimately benefiting patient care and easing clinician workload.

## 8. Conclusion

This study provides a rigorous and standardized assessment of contemporary machine learning algorithms for seizure detection in long-term EMU scalp EEG, utilizing the Sz-CORE framework for transparent and comparable evaluation. While results demonstrated notable advancements and the competitiveness of leading algorithms against commercial software, they also highlighted the persistent difficulties in achieving high sensitivity and precision simultaneously, and starkly revealed the generalization gap between self-reported efficacy and performance on unseen, diverse clinical data.

More significantly than the snapshot of current performance, this challenge transitions into a continuously open benchmarking platform. This initiative, a core outcome of our work, establishes a lasting resource for the machine learning and epilepsy research communities. By providing ongoing access to a standardized evaluation pipeline on a private, expertly annotated dataset, the platform directly addresses the critical need for reproducible research, independent validation, and continuous tracking of progress in the field. It offers a means to objectively compare novel approaches, iterate on existing models, and reduce the discrepancy often observed in self-reported results.

This open benchmark is designed to foster sustained innovation beyond the timeframe of a single competition. It will enable researchers to consistently test their algorithms, allow clinical centers to evaluate solutions on their own data using a standardized methodology, and collectively accelerate the development of robust, generalizable seizure detection models. Ultimately, by transforming this competitive event into a persistent community resource, we aim to catalyze the progress towards clinically impactful machine learning tools that can meaningfully improve patient care in epilepsy.

## Impact Statement

This work aims to advance the field of Machine Learning applied to healthcare, specifically by auditing the reliability of automated seizure detection systems. While the primary societal goal is to improve the diagnosis and monitoring of epilepsy, a condition affecting over 50 million people worldwide, our findings highlight critical ethical and safety considerations for the deployment of AI in clinical settings.

Our identification of a unstable reliability of high-capacity models which achieve peak performance but exhibit instability on outlier patients has direct safety implications. In a clinical context, a model that performs exceptionally well on average but fails catastrophically on specific patients poses a risk of missed diagnoses or safety-critical failures (e.g., SUDEP risk monitoring). This study cautions against the blind adoption of state-of-the-art architectures based solely on aggregate metrics, advocating instead for rigorous stability analysis before clinical deployment.

We expose a pervasive generalization gap between self-reported and actual performance. Addressing this is an ethical imperative; deploying models that fail to generalize leads to inequitable healthcare outcomes, particularly for patient populations underrepresented in public training data (e.g., pediatric cases or those with rare seizure semiologies). Our open benchmarking platform aims to increase transparency and foster the development of models that are robust across diverse patient demographics.

Finally, the open benchmarking platform that we release and keep online as a public resource has the potential to collectively accelerate the development of robust, generalizable seizure detection models.

## Acknowledgement

The competition was supported by Ceribell™ with cash prizes for the two best-performing algorithms. Ceribell waived all organizational control and did not claim intellectual property. The organizing committee of the International Conference on Artificial Intelligence in Epilepsy and Neurological Disorders provided communication support. The EPFL Research Computing Platform provided timely and technically impeccable support to evaluate all algorithms in two weeks. Sándor Beniczky led the data collection effort. Radu Gătej from BrainCapture provided support in parsing the NicoletOne™ EEG files. We also congratulate and thank all the algorithm developers who submitted exciting new algorithms to this challenge.

This work was supported in part by the Swiss NSF, grant no. 10.002.812: "Edge-Companions: Hardware/Software Co-Optimization Toward Energy-Minimal Health Monitoring at the Edge," and in part by the Wyss Center for Bio and Neuro Engineering through the project Lighthouse Noninvasive Neuromodulation of Subcortical Structures.

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

# A. Algorithmic Details

| Algorithm | Method | Input | Pre-processing | | | | Post-proc. | Reported F1 [%] | Code |
| --- | --- | --- | --- | --- | --- | --- | --- | --- | --- |
| | | | Notch | Bandpass | Resample | Features | | | |
| bRAWlstar | Linear Recurrent Unit | – | 60 | ✗ | ✗ | ✗ | – | 29–45 | ✗ |
| Channel-adaptive | CNN | ✗ | ✗ | ✗ | ✗ | ✗ | ✓ | 78 | C |
| ConvNet | CNN | 10 | ✗ | 1–45 | ✗ | ✗ | ✗ | 80–90 | ✗ |
| DeepSOZ-HEM | LSTM & Transformer | 600 | ✗ | ✗ | ✗ | ✗ | ✓ | 89 | C |
| DynaSD | LSTM | 1 | 60, 120 | 1–100 | 128 Hz | ✗ | ✓ | ✗ | C |
| eegwavenet | EEGWavenet & CNN | 4 | ✗ | –64 | ✗ | ✗ | – | 2–31 | C |
| eventNet | U-Net & CNN | 120 | ✗ | ✗ | ✗ | ✗ | ✗ | 22–52 | C |
| Gradient Boost | Gradient Boosted Trees | 10 | ✗ | ✗ | ✗ | ✓ | ✓ | 72 | C |
| HySEIZa | Hyena-Hierarchy & CNN | 12 | 50, 60 | 0.5–120 | ✗ | ✓ | ✓ | 66.01 | ✗ |
| NE Illusion 1 | JEPA (Assran et al., 2023) | 5 | 60 | 0.5–124.9 | ✗ | ✓ | ✗ | 82 | ✗ |
| Random Forest | Random Forest | 2 | ✗ | 1–50 | ✗ | ✓ | ✓ | 80 | ✗ |
| S4Seizure | S4 (Gu et al., 2022) | 12 | 50, 60 | 0.5–120 | ✗ | ✓ | ✓ | 62–70 | ✗ |
| SD2025 | LaBraM (Jiang et al., 2024) | 5 | 50 | ✗ | 200 Hz | ✗ | ✗ | 43 | ✗ |
| STORM | BERT (Devlin et al., 2019) | 60 | – | 1–70 | – | – | – | 75.93 | ✗ |
| SeizFormer | CNN & Transformer | 12 | 50, 60 | 0.5–129 | – | ✓ | – | 54.65 | ✗ |
| SeizUnet | U-net & LSTM | 20.48 | ✗ | 0.5–100 | 200 Hz | ✗ | ✓ | 60.6 | C |
| Sz Transformer | CNN & Transformer | – | 1, 60 | 0.5–120 | ✗ | ✓ | – | ✗ | C |
| Band Powers | – | 5 | 50 | 2–40 | – | – | ✓ | 39 | ✗ |
| Van Gogh Detect | CNN & Transformer | 13x10 | – | – | – | – | ✓ | 60 | ✗ |
| zhu-transformer | CNN & Transformer | 25 | – | – | – | – | – | 11–51 | C |

*Table 2.* Algorithmic details for all submissions. Input duration is expressed in seconds. Post-processing involves variations of prediction output smoothing. The reported F1-score is evaluated by the methods' authors on different validation sets, often hold-outs from the training set. Missing information is marked by a –.

## B. Training Datasets

| Method | TUH Seizure | TUH EEG | Siena Scalp | CHB-MIT | SeizeIT2 | HUP | Total |
|---|---|---|---|---|---|---|---|
| bRAWlstar | ✓ | ✗ | ✗ | ✗ | ✗ | ✗ | 1 |
| Channel-adaptive | ✓ | ✗ | ✓ | ✓ | ✗ | ✗ | 3 |
| ConvNet | ✗ | ✗ | ✗ | ✓ | ✗ | ✗ | 1 |
| DeepSOZ-HEM | ✓ | ✗ | ✓ | ✗ | ✗ | ✗ | 2 |
| DynaSD | ✗ | ✗ | ✗ | ✓ | ✗ | ✓ | 2 |
| eegwavenet | ✗ | ✗ | ✗ | ✓ | ✗ | ✗ | 1 |
| eventNet | ✓ | ✗ | ✗ | ✗ | ✗ | ✗ | 1 |
| Gradient Boost | ✓ | ✗ | ✓ | ✗ | ✗ | ✗ | 2 |
| HySEIZa | ✓ | ✗ | ✗ | ✗ | ✗ | ✗ | 1 |
| NE Illusion 1 | ✓ | ✗ | ✗ | ✗ | ✗ | ✗ | 1 |
| Random Forest | ✗ | ✗ | ✗ | ✓ | ✗ | ✗ | 1 |
| S4Seizure | ✓ | ✗ | ✗ | ✗ | ✗ | ✗ | 1 |
| SD2025 | ✓ | ✓ | ✓ | ✗ | ✗ | ✗ | 3 |
| STORM | ✓ | ✓ | ✗ | ✗ | ✗ | ✗ | 2 |
| SeizFormer | ✓ | ✗ | ✗ | ✗ | ✗ | ✗ | 1 |
| SeizUnet | ✓ | ✗ | ✗ | ✗ | ✓ | ✗ | 2 |
| Sz Transformer | ✓ | ✗ | ✓ | ✗ | ✗ | ✗ | 2 |
| Band Powers | ✗ | ✗ | ✓ | ✗ | ✗ | ✗ | 1 |
| Van Gogh Detect | ✓ | ✗ | ✗ | ✓ | ✗ | ✗ | 2 |
| zhu-transformer | ✓ | ✗ | ✗ | ✗ | ✗ | ✗ | 1 |
| **Total** | 15 | 2 | 6 | 6 | 1 | 1 | |

*Table 3.* Training datasets used by each method. The public datasets used were: TUH EEG Seizure Corpus (Shah et al., 2018), TUH EEG corpus (Obeid and Picone, 2016), Siena Scalp EEG Dataset (Detti, 2020; Detti et al., 2020; Goldberger et al., 2000), Short-term SWEC iEEG (Burrello et al., 2019), Physionet CHB-MIT Scalp EEG dataset (Shoeb, 2009; Goldberger et al., 2000), KU Leuven SeizeIT2 (Bhagubai et al., 2025), Intracranial EEG recordings of Seizures from HUP (Bernabei and Litt, 2021).

| Algorithm | GPU [h] | CPU [h] | RAM [GB/h] |
|---|---|---|---|
| Sz Transformer | 1 | 128 | 173 |
| Van Gogh Detect | 10 | 161 | 254 |
| S4Seizure v2 | 10 | 1'168 | 430 |
| DeepSOZ-HEM | 1 | 44 | 74 |
| S4Seizure v1 | 9 | 1'186 | 630 |
| HySEIZa v1 | 4 | 72 | 153 |
| S4Seizure v3 | 9 | 1'170 | 547 |
| zhu-transformer | 85 | 7'952 | 8'708 |
| SeizUnet | 14 | 210 | 510 |
| HySEIZa v2 | 5 | 69 | 177 |
| Channel-adaptive | 25 | 82 | 122 |
| eventNet | 1 | 108 | 161 |
| Gradient Boost v2 | 0 | 6'006 | 86'681 |
| DynaSD | 0 | 966 | 2'116 |
| Random Forest | 0 | 223 | 937 |
| SD2025 | 3 | 172 | 146 |
| bRAWlstar | 3 | 203 | 559 |
| Gradient Boost v1 | 0 | 6'913 | 85'688 |
| SeizFormer | 6 | 61 | 232 |
| Band Powers | 2 | 52 | 124 |
| NE Illusion 1 v1 | 59 | 3'912 | 5'830 |
| Gradient Boost v3 | 0 | 6'152 | 72'787 |
| NE Illusion 1 v3 | 46 | 6'882 | 19'030 |
| STORM | 0 | 48'773 | 104'361 |
| eegwavenet | 1 | 107 | 177 |
| ConvNet | 2 | 3'494 | 12'234 |
| SD2025 v2 | 5 | 188 | 377 |
| NE Illusion 1 v2 | 31 | 1'431 | 715 |
| **Total** | **335** | **97'885** | **403'933** |

*Table 4.* Computing resources

## C. Computing Resources

## D. Benchmark on other datasets

We evaluate the open-source submissions on three large publicly available datasets and one additional private dataset, namely Physionet CHB-MIT v1.0.0 (Shoeb, 2009; Goldberger et al., 2000), Physionet Siena v1.0.0 (Detti, 2020; Detti et al., 2020; Goldberger et al., 2000), TUH Seizure Corpus v2.0.3 (`train`, `dev` and `eval` subsets) and KU Leuven SeizeIT1 v1.0.0 (Vandecasteele et al., 2020; Chatzichristos and Claro Bhagubai, 2023). The analysis shows none of the algorithms can be evaluated on Physionet CHB-MIT as the dataset only provides a bipolar electrode montage which does not comply with SzCORE data format for which algorithms are designed. Algorithms perform similarly on the Filadelfia and SeizeIT1 datasets which are both long-term recordings from the EMU. Datasets which have a higher proportion of seizures relative to recording duration resulting from curation to only include events (TUH Seizure Corpus) or sub-selection of recording files (Physionet Siena) achieve a higher F1-score.

We compare the datasets using Spearman's rank correlation ($\rho$) of per-algorithm $F_1$ rankings across datasets (excluding algorithms trained on the target to prevent inflation) (table 5.

We found that SeizeIT1 has the highest correlation ($\rho = 0.82$) with Filadelfia., and 11/13 algorithms kept their rank within ±2 positions. Continuous EMU datasets correlate most strongly with Filadelfia, reflecting curation bias.

## E. Statistical results

| Dataset | Curation | $\rho$ vs. Filadelfia | $n$ |
|---|---|---|---|
| SeizeIT1 | EMU, continuous (similar to Filadelfia) | 0.82 | 13 |
| Siena | EMU, sub-selected recordings | 0.733 | 14 |
| TUH | events-only | 0.54 | 14 |

*Table 5.* Spearman's rank correlation compared to Filadelfia results.

| | public datasets | | | private datasets | |
|---|---|---|---|---|---|
| **Algorithms** | **chbmit** | **siena** | **tuh** | **seizeit** | **filadelfia** |
| Sz Transformer | | 🚆 | 🚆 | 36 | 32 |
| DeepSOZ-HEM | | 🚆 | 🚆 | 24 | 30 |
| zhu-transformer | | 48 | 🚆 | 11 | 19 |
| eventNet | | 53 | 🚆 | 22 | 14 |
| SeizUnet | | 20 | 🚆 | 32 | 11 |
| Channel-adaptive | 🚆 | 🚆 | 🚆 | 2 | 7 |
| Gradient Boost v2 | | 🚆 | 🚆 | | 7 |
| DynaSD | 🚆 | 17 | 35 | 7 | 6 |
| SD2025 | | 🚆 | 🚆 | 1 | 2 |
| eegwavenet | 🚆 | 13 | 15 | 1 | 2 |

*Table 6.* Event-based F1-score of the open-source submissions of the challenge is evaluated on three public datasets and one extra private datasets. The additional evaluation datasets are Physionet CHB-MIT v1.0.0 (Shoeb, 2009; Goldberger et al., 2000), Physionet Siena v1.0.0 (Detti, 2020; Detti et al., 2020; Goldberger et al., 2000), TUH Seizure Corpus v2.0.3 and KU Leuven SeizeIT1 v1.0.0 (Vandecasteele et al., 2020; Chatzichristos and Claro Bhagubai, 2023). Datasets that were used for training are marked with a 🚆.

| Algorithm | Rank | *p*-value | $p_{corr}$ | **Cliff's** $\delta$ | **Effect Size** |
|---|---|---|---|---|---|
| *HySEIZa v1* | 9.30 | - | - | - | *(Reference)* |
| DeepSOZ-HEM | 9.53 | 0.955 | 1.000 | -0.07 | Negligible |
| HySEIZa v2 | 9.74 | < 0.001 | **< 0.001** | 0.21 | Small |
| S4Seizure v3 | 9.98 | 0.401 | 1.000 | 0.05 | Negligible |
| S4Seizure v1 | 10.36 | 0.780 | 1.000 | 0.01 | Negligible |
| Van Gogh Detect | 11.19 | 0.955 | 1.000 | -0.01 | Negligible |
| eventNet | 11.34 | < 0.001 | **0.002** | 0.22 | Small |
| SeizFormer | 11.92 | < 0.001 | **< 0.001** | 0.43 | Medium |
| SeizureTransformer | 12.00 | 0.954 | 1.000 | 0.01 | Negligible |
| DynSD | 12.35 | < 0.001 | **< 0.001** | 0.42 | Medium |
| S4Seizure v2 | 12.59 | 0.733 | 1.000 | 0.08 | Negligible |
| zhu-transformer | 12.75 | 0.030 | 0.210 | 0.20 | Small |
| SD2025 v2 | 12.96 | < 0.001 | **< 0.001** | 0.42 | Medium |
| bRAWlstar | 14.07 | < 0.001 | **< 0.001** | 0.44 | Medium |
| ConvNet | 14.51 | < 0.001 | **< 0.001** | 0.48 | Large |
| NE Illusion 1 v1 | 14.61 | < 0.001 | **< 0.001** | 0.46 | Medium |
| eegwavenet | 14.72 | < 0.001 | **< 0.001** | 0.45 | Medium |
| NE Illusion 1 v3 | 15.22 | < 0.001 | **< 0.001** | 0.45 | Medium |
| Serial Band Powers | 15.45 | < 0.001 | **< 0.001** | 0.48 | Large |
| STORM | 15.49 | < 0.001 | **< 0.001** | 0.44 | Medium |
| SD2025 | 15.67 | < 0.001 | **< 0.001** | 0.47 | Medium |
| SeizUnet | 18.24 | < 0.001 | **< 0.001** | 0.44 | Medium |
| Multi-scale Gradient Boosting v2 | 18.65 | < 0.001 | **< 0.001** | 0.52 | Large |
| Multi-scale Gradient Boosting v1 | 18.65 | < 0.001 | **< 0.001** | 0.56 | Large |
| Multi-scale Gradient Boosting v3 | 19.35 | < 0.001 | **< 0.001** | 0.60 | Large |
| Channel-adaptive classifier | 20.87 | < 0.001 | **< 0.001** | 0.59 | Large |
| Random Forest | 21.37 | < 0.001 | **< 0.001** | 0.62 | Large |
| NE Illusion 1 v2 | 23.12 | < 0.001 | **< 0.001** | 0.74 | Large |

*Table 7.* Statistical comparison against the top-ranked algorithm (HySEIZa v1). Omnibus Friedman test: $\chi^2 = 433.55$, $p = 7.040e - 75$ (Significant). Ranks are calculated using the average rank method (lower is better).

