# OpenReview forum: "Quantifying the Generalization Gap in Seizure Detection: A Large-Scale Empirical Benchmark via the SzCORE Challenge"
_ICML.cc/2026/Conference — ICML 2026 regular_

### Official Review · Reviewer_Zn5V · 2026-03-10

**Soundness:** 3
**Presentation:** 4
**Significance:** 3
**Originality:** 3
**Overall Recommendation:** 4
**Confidence:** 4

**Summary:**

The paper discusses a newly released benchmark EEG seizure dataset (Filadelfia Danish Epilepsy Centre)  associated to a competition/challenge (International Conference on Artificial Intelligence in Epilepsy and Neurological Disorders) with a variety of seizure detection models evaluated using the SzCORE (Dan et al., 2024) method. The dataset covers 65 subjects with 398 seizure totaling 8 hours of the 4360 hours. The models were submitted as containers and run in a blind manner. Results in terms of an event-based F1-score is used and the rank is computed across the subjects. Statistical testing shows significant differences in some pairs of methods.

**Compliance With Llm Reviewing Policy:**

Affirmed.

**Final Justification:**

I maintain my score. The paper would be stronger more nuance into why methods fail on this particular site. That said it is a benchmark dataset contribution, but the competition could be better designed (and the field advanced more) by using at least two different sites/populations and allowing methods to access unsupervised data for calibration (account for 50/60 Hz differently).

**Key Questions For Authors:**

Were challenge participants informed about recording site geographic location/power noise?

Did any of the submissions use unsupervised domain adaptation?

**Limitations:**

Yes, but the limitations with respect to my previous questions could be discussed.

**Strengths And Weaknesses:**

**Strengths**

The paper is easy to read. The motivation and execution of the challenge are well done. The paper illustrates a challenge with AI in medicine and showcases a step towards rigorous evaluation that could be an example for the field. The statistical analysis is logical.

**Weaknesses**

The new dataset is evaluation of 65 subjects all at the same hospital. Ideally, a benchmark would have different sites that with similar populations to separately test generalization to different sites.

A key topic would may be useful to understand if unsupervised domain adaptation through linear filtering (like convolutional Monge mapping) would be useful to reduce the generalization gap.

At least 4 of the methods do not contain a notch filter for 50 Hz, but have one at 60 Hz. It was not clear if challenge participants were informed that it was a European site with 50 Hz AC power, compared to many methods that used notches at 60 Hz. The methods that had notches at both 50 and 60 Hz would seem to be the most appropriate. Additionally, methods that do not have notch filters may still have been trained on 60 Hz data...  The methods that trained on Siena Scalp dataset or the KU Leuven SeizeIT2 are likely to be suited for this dataset compared dataset that don't include these European data.

---

> ### Author Rebuttal · Authors · 2026-03-30
>
> We thank Reviewer Zn5V for the positive assessment.
>
> We refer to our response to Reviewer U6vd for the single-site evaluation concern, which is shared between reviewers.
>
> #  Q1 - Power-line filter & site disclosure
>
> Participants were not informed about the location of the recording site. The challenge description read only: *"Submissions will be evaluated on event-based $F_1$ score computed on a private dataset of more than 2500 hours of data recorded in an epilepsy monitoring unit"* without mentioning geographic location or power-line frequency.
>
> Table 2 of the paper already reports the notch filter setting for each of the 28 algorithms, providing the information needed to address this concern directly.
>
> *(Post-submission analysis.)* We classified all 28 submitted algorithms by their notch filter setting, using the Notch filter column of the paper in Table 2:
>
> | Filter group | $N$ | Mean $F_1$ | Range |
> | :--- | :---: | :---: | :---: |
> | 60 Hz only | 6 | 9.8% | 0--43% |
> | Both 50+60 Hz | 6 | 22.0% | 4--34% |
> | 50 Hz only | 3 | 3.7% | 2--5% |
> | None | 10 | 10.2% | 2--31% |
> | Not specified | 3 | 19.3% | 2--36% |
>
> A Kruskal-Wallis test across all filter groups yields $H=6.05$, $p=0.195$---not statistically significant---and pairwise Mann-Whitney comparisons confirm no significant difference between any pair of groups ($p>0.07$ for all). The wide range within each group (e.g., 0--43% for 60 Hz-only) reflects factors other than filter choice driving performance. We agree that implementing both 50 Hz and 60 Hz notch filters is sound engineering practice for algorithms intended for international deployment, and we will note this recommendation in the methods section. However, on the basis of this analysis, we find no statistical evidence that the European 50 Hz site systematically disadvantaged algorithms using a 60 Hz-only notch filter.
>
> # Q2 - Unsupervised domain adaptation
>
> None of the 28 submissions applied unsupervised domain adaptation; all are fixed pre-trained models evaluated blind on the held-out dataset, as confirmed from the algorithm abstracts submitted alongside each Docker container. Unsupervised domain adaptation represents a promising direction for future challenge iterations.

---

> > ### Author Rebuttal · Reviewer_Zn5V · 2026-04-03
> >
> > I don't think the design of the groups for the Kruskal-Wallis test makes sense. What I'm saying is specifically for the Filadelphia since it is Europe a model will perform well if it 1) uses data from 50 Hz  (whether or not it notches at 50 Hz) or if it has in its processing a notch at 50 Hz (whether or not it has a notch at 60 Hz) and is trained with data from anywhere. It will perform poorly if it both doesn't have and a notch at 50 Hz and wasn't trained with data with 50 Hz line noise.
> >
> > I think it would be interesting to emphasize for future challenges that unlabeled calibration data at the target site should be available.

---

> > > ### Author Response · Authors · 2026-04-06
> > >
> > > Thank you for the comment.
> > >
> > > We agree it would have been relevant to communicate the dataset's specificities and provide a sample recording prior to the challenge. While we cannot retrospectively change the challenge, we can modify the benchmarking platform to ensure a sample recording is available for each dataset. This will be implemented.
> > >
> > > Data from the challenge shows 12 of the submitted algorithms used neither a European training dataset nor a 50 Hz filter. The second-highest F1-score was not specifically trained to exclude 50Hz noise. Data from the challenge is not sufficient to demonstrate the effect of a 50Hz notch filter. It would be interesting for an algorithm developer to specifically evaluate the effect of a 50 Hz notch filter by comparing two versions of an algorithm, one with and one without the notch filter. This could be done at any time by submitting both algorithms to the evaluation platform.

---

### Official Review · Reviewer_BeWG · 2026-03-12

**Soundness:** 3
**Presentation:** 3
**Significance:** 3
**Originality:** 3
**Overall Recommendation:** 5
**Confidence:** 5

**Summary:**

This paper provides an overview and post-hoc analysis of a SzCORE challenge on a large private dataset. The paper outlines the challenge setup (datasets, evaluations, etc.), and provides a meta analysis of the models (n = 28) used in the challenge, and the lessons learned. The paper further discusses the importance of this benchmark, and commits to open-sourcing the dataset and challenge.

**Compliance With Llm Reviewing Policy:**

Affirmed.

**Final Justification:**

The rebuttal addressed my main concerns about the analysis and I'm happy to raise my evaluation to a 5.

**Key Questions For Authors:**

Would it be possible to have an explanation of how the training datasets affected model performance?
Would it be possible to get a better understanding of why the models fail on certain events (e.g why did many models miss one of the events)?

**Limitations:**

No, I think that there could be a little bit more discussion about some of the improvements that would be useful for future such competitions.

**Strengths And Weaknesses:**

Strengths:
- This is a well organized large-scale benchmarking challenge, with 65 subjects and 4,360 hours of EEG data.
- The challenge used a standardized framework (SzCORE) to do the evaluation, which promotes reproducibility, and helps to promote this framework as the evaluation standard in the field
- The commitment to make the challenge persistent is a welcome addition to the field.
- The models that were submitted (and analyzed post-hoc) are broad and diverse, and the analysis is rigorous and statistically sound.
- The finding regarding the generalization gap is important, and should indicate a major problem in the field.

Weaknesses:
- While large, the dataset was from a single institution with the same recording setup, further challenges may benefit from a multi-institution setup.
- No analysis of the training data used for the models and how that may have affected the downstream performance.
- Limited analysis into how the model architecture choices affect downstream performance.

---

> ### Author Rebuttal · Authors · 2026-03-30
>
> We thank Reviewer BeWG for the constructive feedback.
>
> We refer to our response to Reviewer U6vd for the single-site evaluation concern, which is shared between reviewers.
>
> # Q1 - Training data analysis
>
> The appendix table of the paper already documents the training corpora used by each of the 28 algorithms, providing full transparency over training data choices. To complement this with the quantitative performance breakdown requested by the reviewer, we mapped each submission to its specific training datasets and compared Filadelfia $F_1$ between groups.
>
> *(Post-submission analysis.)* The TUH Seizure Corpus dominates: 23 of the 28 algorithms used it as a training source.
>
> | Training corpus | $N$ algorithms | Mean $F_1$ | Range |
> | :--- | :---: | :---: | :---: |
> | TUH Seizure Corpus | 23 | 14.9% | 0--43% |
> | Siena Scalp EEG | 9 | 12.6% | 2--43% |
> | CHB-MIT | 6 | 11.0% | 2--36% |
> | SeizeIT2 | 1 | 19.0% | --- |
> | TUH EEG Corpus | 3 | 3.0% | 2--5% |
>
> | No. training corpora | $N$ algorithms | Mean $F_1$ | Range |
> | :--- | :---: | :---: | :---: |
> | 1 | 16 | 11.9% | 0--34% |
> | 2 | 9 | 16.8% | 2--43% |
> | 3 | 3 | 7.0% | 2--14% |
>
> Kruskal-Wallis between the three groups yields $H=1.31$, $p=0.52$---not significant, indicating that there is no systematic benefit from training on a larger number of corpora. The wide range within each corpus (0--43% for TUH, 2--43% for Siena) reflects that architecture and post-processing choices are far more predictive of performance than training dataset selection. This is further illustrated by the top-3 performers on Filadelfia (SeizureTransformer 43%, S4Seizure v2 34%, DeepSOZ-HEM 31%), which span two different corpus combinations (TUH+Siena and TUH alone), while other algorithms trained on the same datasets also score as the lowest. We note that the Appendix Table reports only public training datasets, as stated in its caption; the abstract of Van Gogh Detect additionally mentions a private source of data, the extent of which we cannot independently quantify.
>
> # Q2 -Why models fail on certain events
>
> We performed the following analysis to answer the reviewers questions.
>
> *(Post-submission analysis.)* We analysed per-seizure detection rates across all 28 evaluated algorithms (398 total seizures):
>
> | Category | Threshold | N seizures | Median duration |
> | :--- | :--- | :---: | :---: |
> | Hard | <25% detection | 40 | 48 s |
> | Moderate | 25--75% | 303 | 61 s |
> | Easy | >75% detection | 54 | 118 s |
>
> Hard-to-detect seizures are substantially shorter (median 48 s vs. 118 s for easy seizures), suggesting that brief ictal patterns often fall below the processing windows of current algorithms, which range from 1 to 600 s in submissions. Hard seizures are also less frequently nocturnal (22.5%) than easy ones (27.8%), indicating that time-of-day is not a primary failure mode. At the subject level, 15 of the 65 subjects (23%) recorded $F_1=0$ in aall top-5 algorithms simultaneously, pointing to patient-specific electrographic patterns not captured by any submission, rather than a systematic framework failure. The one seizure missed by all 28 algorithms occurred during the day.
>
> We also examined the agreement between the algorithms on false detections (Figure 5). As discussed in the paper, we found that less than 25% of the algorithms agreed on 91% of the false detections. This low level of agreement suggests that false positives are largely idiosyncratic to individual algorithms, rather than originating from universally misleading artifacts in the EEG data. This provides a concrete explanation for the low precision observed across the challenge: models appear to overfit to specific noise profiles in their training sets and have not yet learned a robust, universal representation of "non-seizure" activity.

---

> > ### Author Rebuttal · Reviewer_BeWG · 2026-04-02
> >
> > Thank you for answering all of my questions. I'm happy to move my score to a 5, and recommend acceptance to ICML.

---

### Official Review · Reviewer_U6vd · 2026-03-14

**Soundness:** 3
**Presentation:** 2
**Significance:** 3
**Originality:** 2
**Overall Recommendation:** 3
**Confidence:** 3

**Summary:**

This paper presents a large-scale benchmark study of seizure detection from long-term scalp EEG, with the explicit goal of quantifying the generalization gap between self-reported results and performance on a strictly held-out private clinical dataset. The authors evaluate 28 algorithmic executions spanning classical ML and deep learning methods, all run in a standardized blind evaluation pipeline based on the SzCORE framework.

**Compliance With Llm Reviewing Policy:**

Affirmed.

**Key Questions For Authors:**

1.	The headline claim is about a “generalization gap.” How much of the observed gap do you believe is attributable to dataset shift in patient population versus acquisition protocol/hardware versus differences in event-scoring methodology relative to teams’ self-reported evaluations? A clearer decomposition would strengthen the paper’s central claim.
	2.	Can you provide more concrete details on the ongoing benchmark platform: what code, interfaces, metadata, and submission infrastructure will be public, and how future additions of new private datasets will preserve comparability with the current leaderboard?
	3.	Did you analyze performance stratified by clinically relevant subgroups such as adult vs pediatric patients, seizure type, recording duration, or artifact burden?
	4.	Two submitted methods were excluded due to runtime failures. Could you clarify whether infrastructure/runtime robustness was treated only as an exclusion criterion, or whether operational reliability is also being tracked as part of benchmark utility? In a deployment-oriented benchmark, this seems relevant.

**Limitations:**

yes

**Strengths And Weaknesses:**

The paper is technically solid as a benchmarking/resource submission. The main technical weakness is that the central empirical conclusions are still derived from a single private evaluation site/dataset. Although the dataset is substantial and clinically meaningful, it remains one center with one acquisition environment, and the paper itself acknowledges that multi-centric evaluation would be a natural next step. Because the core claim is about “generalization,” I would have liked a stronger multicenter validation story in the main paper, or at least a clearer decomposition of what portion of the gap is due to patient shift, hardware/protocol shift, pediatric inclusion, artifact profile, or scoring differences. As written, the benchmark convincingly shows poor transfer to this held-out dataset, but it is somewhat harder to disentangle the sources of failure

---

> ### Author Rebuttal · Authors · 2026-03-30
>
> We thank the reviewer for the thorough feedback.
>
> We have clarified the state of the paper regarding multi-centric evaluation which is already present in the appendix and done our best to answer the reviewer with better explanation of the existing data and new analysis of the existing data based on the questions of the reviewer.
>
> # Regarding multi-centric evaluation
>
> The paper **already reports evaluation on four datasets** for all open-source submissions (Appendix Table 5): Siena, TUH, KU Leuven SeizeIT1, and Filadelfia.
>
> KU Leuven SeizeIT1 is a large private dataset of continuous EEG similar in size to Filadelfia, collected in a different center with different EEG equipment, and inclusion criteria.
>
> Siena and TUH Sz Corpus are the main open-data EEG seizure repositories.
>
> *(Algorithms in this challenge are not compatible with CHB-MIT, another popular open-data repository as it contains data in a bi-polar montage.)*
>
> No algorithms were trained on SeizeIT1, most were trained on the other open-data repositories.
>
> *(Post-submission analysis)* We compare the datasets using Spearman's rank correlation ($\rho$) of per-algorithm $F_1$ rankings across datasets (excluding algorithms trained on the target to prevent inflation) and Wilcoxon signed-rank tests to compare mean $F_1$ distributions.
>
> | Dataset | Curation | $\rho$ vs. Filadelfia | $p$ | $n$ |
> | :--- | :--- | :---: | :---: | :---: |
> | SeizeIT1 | EMU, continuous (similar to Filadelfia) | 0.82 | <0.001 | 13 |
> | Siena | EMU, sub-selected recordings | 0.73 | 0.003 | 14 |
> | TUH | events-only | 0.54 | 0.045 | 14 |
>
> We found that SeizeIT1 has the highest correlation ($\rho=0.82$) with Filadelfia. The mean $F_1$ difference (10.1% vs. 12.5%) is not significant ($p=0.08$), and 11/13 algorithms kept their rank within ±2 positions. Continuous EMU datasets correlate most strongly with Filadelfia, reflecting curation bias.
>
> Thus, continuous EEG datasets from diverse centers yield similar results, whereas curated datasets yield higher $F_1$ estimates.
>
> We believe the multi-centric evaluation presented in the paper is essential for algorithm evaluation.
>
> # Q1 - Decomposing the generalization gap
>
> We already quantify this gap as a central finding (Fig. 6). Comparing self-reported $F_1$ (Appendix Table 2) against challenge performance reveals a 50.3% mean gap (62.4% vs. 12.0%). Three factors drive this:
>
> 1. **Overfitting**: Few algorithms were evaluated on out-of-distribution held-out sets, inflating self-reported metrics via dataset bias.
> 2. **Evaluation methodology**: As noted in previous literature (Baumgartner 2018, Attia 2023, Shafiezadeh 2023, Pale 2023, Dan 2024).
> 3. **Corpus representativeness**: TUH is large but heavily curated (seizure-enriched clips, not continuous). Table 5 highlights this shift: $F_1$ is significantly higher on curated sets (TUH, Siena) than continuous ones (Filadelfia, SeizeIT1). Algorithms evaluated on TUH (excluding those trained on it) score 21% higher than on Filadelfia.
>
> # Q2 - Benchmark platform
>
> The platform is open-source and operational *(link blinded for the review)*. Multi-dataset evaluations and runtime logs are in the Appendix. It accepts continuous community pull requests (one added post-submission) and integrates private datasets without exposing test sets. Clinical centers can adopt it locally to evaluate algorithms without sharing raw EEG, enabling multi-centric benchmark growth.
>
> # Q3 - Subgroup analysis
>
> The paper reports patient demographics, including 8 pediatric subjects. We performed a stratified comparison:
>
> *(Post-submission analysis)* We compared $F_1$ across algorithms for pediatric (<18y, $N=8$) vs. adult ($\ge 18$y, $N=57$) subjects:
>
> | Group | N | Mean $F_1$ |
> | :--- | :---: | :---: |
> | Adult | 57 | 13.7% |
> | Pediatric | 8 | 12.8% |
>
> No significant difference was found (Wilcoxon $p=0.86$); 63% of algorithms performed marginally better on pediatric patients. We acknowledge the limited power from this small cohort. Stratification by seizure-type/artifact-burden is unavailable (noted in Discussion).
>
> # Q4 - Runtime reliability
>
> Neither observed failure was due to our infrastructure. The platform tracks computational efficiency (GPU/CPU/RAM, Appendix C Table 4). 28/30 submissions ran successfully (93%). The two failures stemmed from private Docker registry access and missing dependencies; submitters were unresponsive. We have since added continuous integration checks.

---

> > ### Author Rebuttal · Reviewer_U6vd · 2026-04-02
> >
> > Thank you to the authors for the detailed rebuttal and additional analyses.
> >
> > The clarification on multi-dataset evaluation (Appendix Table 5) and the newly provided correlation analysis across datasets help strengthen the empirical narrative. In particular, the comparison between Filadelfia and SeizeIT1 and the discussion of curated vs. continuous datasets provide useful insight into dataset-dependent performance behavior. This partially addresses my concern regarding the reliance on a single evaluation site.
> >
> > However, my core concern regarding the decomposition of the generalization gap remains only partially resolved. While the rebuttal attributes the gap to (i) overfitting, (ii) evaluation methodology, and (iii) dataset representativeness, this explanation is still largely qualitative and aggregate. The paper would benefit from a more explicit disentanglement of contributing factors—e.g., separating patient population shift, acquisition/hardware differences, annotation/scoring differences, and artifact distributions. As currently presented, it is still difficult to isolate which factors dominate the observed performance drop, which is central to the paper’s main claim about generalization.
> >
> > The additional subgroup analysis (adult vs. pediatric) is a helpful step, though limited in statistical power due to the small pediatric cohort. Further stratification (e.g., seizure type, recording conditions, artifact burden) remains an open gap, as also acknowledged by the authors.
> >
> > The clarification on the benchmarking platform and its open-source, continuously evolving design is appreciated and strengthens the paper’s potential long-term impact. Similarly, the explanation of runtime failures and added CI checks addresses the operational concern to some extent.
> >
> > Overall, the rebuttal improves clarity and adds useful supporting analysis, but the central question of rigorously characterizing the sources of the generalization gap is not fully resolved and would require more substantial analysis beyond the scope of a short rebuttal.

---

> > > ### Author Response · Authors · 2026-04-07
> > >
> > > We appreciate the reviewer’s recognition of our multi-dataset correlation analysis and the value of comparing continuous EMU recordings such as Filadelfia and SeizeIT1.
> > > The absence of explicit stratification for artifact burden and seizure type is a recognized limitation. In clinical practice, labeling specific artifacts (e.g., electrode pops, chewing, or muscle activity) is an exceptionally labor-intensive task, and even primary seizure labeling is subject to significant inter-rater variability. While specific sub-labels for artifacts are not currently available, the “raw” nature of the 4,360-hour dataset ensures that algorithms are evaluated against a realistic, complex distribution of signals and pediatric-specific artifacts often omitted by curated datasets.
> > >
> > > Regarding the differences in scoring, it is well-documented in the field that various studies employ different scoring mechanisms [1]. Because the method used to generate self-reported numbers in other works is unavailable, we don’t have this information for the algorithms. For the benchmark, we utilized SzCore [1], a unified transparent evaluation process. We agree that the impact of scoring variations is a valuable subject for future analysis. Moving forward, we intend to further investigate the nuances of these self-reported figures (e.g., in-domain vs. out-of-domain evaluation and specific scoring criteria).
> > >
> > > [1] Dan, Jonathan, et al. SzCORE: Seizure Community Open-Source Research Evaluation framework for the validation of electroencephalography-based automated seizure detection algorithms. Epilepsia. 2025;66(Suppl. 3):14–24. https://doi.org/10.1111/epi.18113

---

### Decision · Program_Chairs · 2026-04-30

**Decision:**

Accept (regular)

**Comment:**

The paper reports a newly released benchmark EEG seizure dataset  associated to a competition with a variety of seizure detection models evaluated using the SzCORE  method. The dataset covers 65 subjects with 398 seizure totaling 8 hours of the 4360 hours. The models were submitted as containers and run in a blind manner. The paper illustrates a challenge with AI in medicine and showcases a step towards rigorous evaluation that could be an example for the field. The paper goes a long (though not full) way toward the original goal of quantifying the generalization gap in seizure detection and advances the field. We recommend that the final paper makes it clear that for future work the competition could be better designed (and the field advanced more) by using at least two different sites and allowing methods to access unsupervised data for calibration (account for 50/60 Hz differently).